# LLM Hallucination Detection: Fine-Tuning Gemma2

**Ziyi Liu**
Department of Electronic Engineering
Tsinghua University
liuziyi24@mails.tsinghua.edu.cn

**Rosalie Butte**
Department of Computer Science and Technology
Tsinghua University
btlsl24@mails.tsinghua.edu.cn

## 1 Introduction

Large Language Models (LLMs) have become more and more important recently in our daily lifes, for example by using ChatGPT, Llama, chatGLM and Claud. However, these models are not without errors and can sometimes give false or misleading answers, called hallucinations. The general definition of hallucination refers to generated content that is nonsensical or unfaithful to the input source content [Ji et al., 2023]. Hallucinations are primarily caused by two factors: discrepancies between the source and reference training data, and inherent model design flaws, such as the encoder learning imperfect representations, decoding strategies introducing randomness, prioritization of pre-trained parameters over input data, or error accumulation during sequence generation based on previously generated tokens. Hallucinated content often seems fluent and natural. Depending on the context, users may not easily detect such errors, which can lead to serious consequences, such as reduced model performance and potential safety issues. Therefore, it is of great importance to develop a method to detect hallucinations in LLMs.

Hallucination detection has become a key area of focus. Moreover, with the growing popularity of LLMs, their strong reasoning capabilities can now be leveraged to detect hallucinations in LLM-generated content. For instance, GPTScore [Fu et al., 2023], SelfCheckGPT [Manakul et al., 2023], ChatProtect [Mündler et al., 2024] and so on.

In this project, we propose a model to help detect the hallucinations in LLMs.

## 2 Problem Description

We orient ourselves at the Kaggle Competition: "ML Olympiad - Detect hallucinations in LLMs" [Massaron, 2024]. The competition provides a large-scale training and a testing set. Both sets contain prompts and their answers. Additionally, the training set contains a label to indicate whether or not the answer is a hallucination. The training set contains 16,687 records and testing set contains 11,126 records. The prompts are from the Open-Orca dataset and were executed using a Mistral 7B Instruct large language model. With this data, we aim to develop a framework that can accurately distinguish between hallucinations and correct answers of LLMs. To achieve this we plan to train a model by fine-tuning Gemma2.

We define this problem as a binary classification task. Given a prompt-response pair $\mathcal{P} = \{\mathcal{Q}, \mathcal{A}\}$, our goal is to train a model $\mathcal{M}$ that can accurately identify whether the response $\mathcal{A}$ contains hallucinations. The model output is $Y = \mathcal{M}(\mathcal{P})$, where $Y \in \{0, 1\}$. $y_i = 1$ represents $\mathcal{A}$ contains a hallucination, and $y_i = 0$ refers to no hallucination.

First, we split the original prompt-response pair into a set of question-answer pairs based on Chain-of-Thought (CoT). Then we utilize these intermediate pairs to fine-tune Gemma2 to classify them.

37th Conference on Neural Information Processing Systems (NeurIPS 2023).

To evaluate our model we will compare its accuracy with several other established models, such as SelfCheckGPT [Manakul et al., 2023], Chainpoll [Friel and Sanyal, 2023] and ChatProtect [Mündler et al., 2024].

## 3 Related Work

In general conversational scenarios, current LLM hallucination detection techniques fall into two main categories: external knowledge retrieval and self-evaluation. These approaches are then combined with parameter-based methods or prompt engineering techniques, such as Chain-of-Thought (CoT) [Wei et al., 2022].

In the first category, the models utilize external knowledge to assist in detection. [Gou et al., 2024] proposed CRITIC, which leverages external tools like search engines to verify LLM outputs and uses CoT to enhance the output.

In the second category, self-evaluation involves using an LLM to detect hallucinations either generated by itself or by another LLM. [Fu et al., 2023] leverages a pre-trained LLM to evaluate generated text based on task-specific instructions.[Manakul et al., 2023] proposed SelfCheckGPT, the first zero-reference factual hallucination detection framework, which measures consistency between the target response and various generated samples using five parameter-based methods.

[Luo et al., 2023] combines prompt engineering techniques into self-evaluation and proposes Self-Familiarity, which utilizes concept extraction and a guessing method to detect instructions that may lead to hallucinations. In which case, it does not generate a response, to reduce the risk of hallucinations. This method differs from others as it checks for the risk of a hallucination before generating a response, contrary to most methods, which focus on detecting hallucinations after the response has already been generated.

[Friel and Sanyal, 2023] treats hallucination detection as a binary classification task, where the LLM identifies hallucinations in text and uses CoT reasoning to justify its classification and provide explanations.

As existent metrics perform differently across various tasks, such as question and answering, fact checking or summarizing, to classify hallucinations, [Valentin et al., 2024] proposes a multi-scoring system. This method first calculates multiple scores using various methods, calibrates them and lastly combines them to achieve a more accurate overall-score indicating if a response is a hallucination or not.

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
