# OpenReview forum: "[Proposal-ML] LLM Hallucination Detection: Fine-Tuning Gemma2"
_tsinghua.edu.cn/THU/2024/Fall/AML — THU 2024 Fall AML Submission_

### Official Review · ~Xiying_Huang2 · 2024-11-09
**Fine-Tuning Gemma2 for Effective Hallucination Detection in LLMs**

**Rating:** 9
**Confidence:** 4

**Review:**

This proposal presents an innovative approach to detect hallucinations in large language models (LLMs) by fine-tuning the Gemma2 model. The project builds on recent advancements in hallucination detection, aiming to classify LLM-generated responses as either hallucinated or accurate using a binary classification framework. The methodology involves leveraging Chain-of-Thought (CoT) reasoning to split prompt-response pairs into intermediate question-answer segments, improving detection precision. The paper is clear, technically sound, and well-grounded in relevant literature, presenting meaningful insights into a critical area of NLP.

Pros:
	•	Addresses a significant and timely issue in LLM development.
	•	Incorporates a structured approach using CoT and Gemma2 fine-tuning.
	•	Compares accuracy with other established hallucination detection models.

Cons:
	•	Lack of clarity on evaluation metrics beyond accuracy.
	•	Potential challenges in CoT application for varied LLM prompt-response formats.

---

### Official Review · ~Jinsong_Xiao1 · 2024-11-09
**review for proposal 27**

**Rating:** 8
**Confidence:** 4

**Review:**

The paper proposes a method to detect hallucinations in large language models (LLMs) by fine-tuning the Gemma2 model with Chain-of-Thought (CoT) reasoning. The method targets a binary classification problem to distinguish between hallucinated and factual responses, with comparisons planned against established models like SelfCheckGPT and ChatProtect. This work is situated within the growing field of LLM reliability.

strength:

- Relevance: Addresses a timely problem as LLMs see increased deployment.

- Clarity: Well-organized with a clear approach and solid literature review.

- Adequate research on related work.

Weakness

- As a binary classification problem seems to be something unnovel, despite the use of large model fine-tuning and CoT and other technologies.

---

### Official Review · ~Yuanda_Zhang1 · 2024-11-09
**review for proposal 27**

**Rating:** 8
**Confidence:** 4

**Review:**

The proposal presents a focused effort on detecting and mitigating hallucinations in Large Language Models (LLMs) by fine-tuning the Gemma2 model. The project aims to address the critical issue of inaccuracies in LLM outputs, which can lead to significant consequences.

Pros:
1）The project's relevance is highlighted by the increasing dependence on LLMs and the pressing need for reliability in their outputs.
2）The proposal's approach to use Gemma2, a well-established model, as a foundation for fine-tuning shows promise in improving detection capabilities.
3）The binary classification task definition is clear and provides a straightforward framework for evaluating the model's performance.

Cons:
1）The proposal could benefit from a more detailed explanation of how the model will handle the nuances of different types of hallucinations, beyond just identifying them.
2）While the project plans to compare its model with established ones, there is a lack of preliminary results or benchmarks to suggest the expected improvement over current methods.

---

### Official Review · ~Tim_Bakkenes1 · 2024-11-09
**Good proposal**

**Rating:** 8
**Confidence:** 3

**Review:**

This is a good proposal.

- The background is relevant and motivates the need for your research well.
- The problem description is and it is good that you describe the datasets you will be using and give a formal problem definition.
- You provide some examples of other models that will be used to evaluate the performance of your model but more motivation for your choice of models and some information about how the comparison would work would improve your proposal.
- The related work is relevant and could have been used to come up with more ideas on your method.
- While the related work section is good, it would have been nice if you used that research to come up with a method tailored to your competition for achieving high accuracy.

---

### Official Review · ~Renrui_Tian1 · 2024-11-10
**Clear Problem Definition, but Methodology and Challenges Need More Specificity**

**Rating:** 7
**Confidence:** 3

**Review:**

**Strengths**:
* **Clear Problem Definition**: The proposal effectively defines the problem of LLM hallucination detection and its significance. It highlights the risks associated with hallucinations and the need for accurate detection methods.
* **Related Work**: The proposal includes a comprehensive survey on the relevant techniques, demonstrating a good understanding of the existing literature on LLM hallucination detection.

**Areas for Improvement**:
* **Detailed Fine-tuning Approach**: The proposal could benefit from a more detailed description of the fine-tuning process. This includes specifics about the training regimen, hyperparameter tuning, and the use of Chain-of-Thought (CoT) for splitting prompts and responses.
* **Challenges**: The proposal falls short in presenting the potential challenges to overcome when training a binary classification model to detect hallucination.

**Overall, this proposal presents a well-structured and promising approach to LLM hallucination detection. Addressing the suggested areas for improvement would further strengthen the proposal and increase the likelihood of successful implementation**.

---

### Official Review · ~Fabian_Pawelczyk1 · 2024-11-11
**Important Topic, Clear Introduction but lack in Methodology**

**Rating:** 8
**Confidence:** 4

**Review:**

# Decision: Clear Accept

## Strengths

- **Problem Significance**: This paper addresses hallucination detection in Large Language Models (LLMs), a critical and timely issue given the increasing use of LLMs in applications that require high factual accuracy.
- **Clarity and Organization**: The paper is clearly organized, with well-defined sections for problem description, methodology, and related work, making it easy to follow.

## Areas for Improvement

- **Problem Definition**: While the introduction discusses the importance of hallucination detection, it could go deeper into explaining why hallucination is a particularly challenging problem.
- **Pipeline Details**: Providing more specifics on the fine-tuning approach for Gemma2 would enhance the methodology section and clarify the technical contributions.

---

Overall, this is a strong and well-prepared proposal addressing an important topic.

---

### Official Review · ~Cheng_Gao2 · 2024-11-11
**Review for LLM Hallucination Detection: Fine-Tuning Gemma2**

**Rating:** 8
**Confidence:** 4

**Review:**

Strengths:

- Clear task definition.
- Hallucination detection is a research hotspot and holds great importance.

Weaknesses:

- The proposed method seems a little bit simple. I think the trained Gemma2 model may rely heavily on its pre-existing knowledge to judge the accuracy of a response, rather than learning new knowledge from the approximately 10,000 training samples. Thus, I am concerned that the trained Gemma2 may not perform significantly better than an untrained version.
- The proposal could benefit from a more detailed description of how the language model is adapted into a binary classification model. Slight architectural adjustments at the language model head may be necessary.

---

### Official Review · ~Ruowen_Zhao1 · 2024-11-11
**Review on LLM Hallucination Detection: Fine-Tuning Gemma2**

**Rating:** 7
**Confidence:** 4

**Review:**

**Summary**

The project focuses on the Kaggle competition "ML Olympiad - Detect hallucinations in LLMs," where the goal is to classify language model responses as hallucinations or correct answers. The authors aim to apply a Chain-of-Thought (CoT) approach to split prompt-response pairs and fine-tune the Gemma2 model for this binary classification task.

**Strengths:**
The approach is presented with good clarity, outlining the dataset, problem definition, and methodology in a structured manner. The related work section is well-developed and clearly articulated, providing a strong foundation for the proposed approach.

**Weakness:**
+ Analysis on model selection: While the authors plan to fine-tune Gemma2, there is limited explanation or justification for selecting this specific model. More detail on why Gemma2 is well-suited for hallucination detection should be provided.
+ Lack of fine-tuning details: There is no specific explanation of how the model will be fine-tuned, such as hyperparameters, optimization strategy or any adjustments to the model architecture.
+ Lack of evaluation metrics: The approach mentions fine-tuning Gemma2 but does not provide a detailed plan for evaluating the model's performance.

---

### Official Review · ~Killian_Conyngham1 · 2024-11-12
**Review of LLM Hallucination Detection: Fine-Tuning Gemma2**

**Rating:** 8
**Confidence:** 4

**Review:**

Overall this is a strong and well-structured proposal.

The introduction provides a clear overview of the problem of Hallucinations, its importance and relevancy. The problem description section expands on this well by clearly introducing the exact classification task involved and the proposed technique of using Chain-of-Thought (CoT). It would be useful to have a more detailed description of how exactly you plan on fine-tuning Gemma2, and to go into more detail in which ways you plan to expand or improve on existing CoT-based detectors. The related work section gives an insightful and detailed overview of the relevant literature. One suggestion would also be to perhaps also compare the hallucination detection accuracy to other non-LLM based approaches, or even manual review for a smaller subset of the data as another reference point.

---

### Official Review · ~Jackson_M_Luckey1 · 2024-11-12
**Proposal Review**

**Rating:** 8
**Confidence:** 4

**Review:**

The proposal does a good job of explaining their interpretation of LLM "hallucinations". While I do not agree with the entirety of the choosen definition, clearly explaining the definiton makes the paper much easier to evaluate. Using a Kaggle dataset with labelled hallucinations is a great approach.

The related work section provides a good overview of the existing literature.

I would like to know more about the methodology you plan on using. I found the proposal sparse on technical details, but the 2 page cap is obviously a limiting factor. I am particularly interested in what chain-of-thought approach you plan on going with.

---

### Official Review · ~Han-Xi_Zhu1 · 2024-11-12
**Review of LLM Hallucination Detection: Fine-Tuning Gemma2**

**Rating:** 8
**Confidence:** 4

**Review:**

This work gives a clear proposal on their topic "Hallucination Detection" with their well organized writing. The authors offer easy-understanding interpretion on the focused problem and give detailed and solid related literature on their topic.

 Their proposed approach is easy to follow. What I concern is if there are any other works dealing with the same problem and how they perform on the specific task. And the authors may have to clarify the reason why they choose Gemma2.

Thank you!

---

### Official Review · ~Wuqian1 · 2024-11-12
**Review of "LLM Hallucination Detection: Fine-Tuning Gemma2"**

**Rating:** 5
**Confidence:** 3

**Review:**

The proposal "LLM Hallucination Detection: Fine-Tuning Gemma2" addresses a critical issue in the field of natural language processing—the problem of hallucinations in Large Language Models (LLMs),The proposal is not given  a clear methodology for training and evaluating the model.
Pros
   1.Relevance: The project addresses a current and pressing issue in AI, making it highly relevant.

Cons
    1.Unproven Effectiveness: The effectiveness of fine-tuning Gemma2 for hallucination detection is yet to be proven.
    2.Methodical:The proposal is not given  a clear methodology for training and evaluating the model.
    3.Lack of Experimental Results: As a proposal, it lacks experimental results to support the claimed potential of the method.